# Back to the Future? The Fallopian Tube, Precursor Escape and a Dualistic Model of High-Grade Serous Carcinogenesis

**DOI:** 10.3390/cancers10120468

**Published:** 2018-11-28

**Authors:** T. Rinda Soong, David L. Kolin, Nathan J. Teschan, Christopher P. Crum

**Affiliations:** 1Department of Pathology, University of Washington Medical Center, Seattle, WA 98195, USA; tsoong@uw.edu; 2Department of Pathology, Division of Women’s and Perinatal Pathology, Brigham and Women’s Hospital, Boston, MA 02115, USA; dkolin@bwh.harvard.edu (D.L.K.); nteschan@bwh.harvard.edu (N.J.T.)

**Keywords:** *TP53*, fallopian tube, high grade serous carcinoma, precursor, p53 signature, SEE-FIM

## Abstract

Beginning with the discovery of the *BRCA*-associated ovarian cancer susceptibility genes and subsequent detailed examination of risk-reducing salpingo-oophorectomy (RRSO) specimens, a new paradigm of ovarian carcinogenesis has unfolded with attention to the distal fallopian tube. The primary focus has been an early cancer or neoplasm in the fallopian tube which is seen in virtually all incidentally discovered high-grade serous cancers in asymptomatic women. This high-frequency of tubal involvement in early serous neoplasm (usually in the form of serous tubal intraepithelial carcinoma—STIC) has galvanized attention to this organ as a primary source of this disease. However, an enduring mystery has been the relatively low frequency of STIC in the fallopian tubes of women with advanced malignancy. This paradox, a high frequency of tubal involvement early on and a low frequency of involvement later in the disease process, has spurred interest in other potential sources, such as the ovarian surface epithelium or cortical inclusions and the secondary Mullerian system. However, because essentially all high-grade serous carcinomas are linked by *TP53* mutations, and because fallopian tubes frequently contain early serous proliferations (ESPs) with these mutations, attention has turned to the possibility that the nonmalignant but *TP53* mutated tubal epithelium could be responsible for an eventual malignancy. Recent data have shown evidence of a lineage continuity between ESPs and concurrent serous carcinomas prompting the concept of “precursor escape”. This creates a second component of the paradigm by which cells from early precursors are shed from the tube and undergo subsequent malignant transformation, emerging suddenly as widespread intraperitoneal malignancy. This dualistic model thus provides a unique pathway by which the future outcome (wide spread high-grade serous carcinomas—HGSC) is ultimately explained by going back in time to an early serous proliferation. This paradigm also brings the peritoneal cavity into focus, raising new questions about the potential co-variables or exposures that might facilitate the occasional malignant transformation of an ESP in the peritoneal cavity or on the peritoneal surface.

## 1. The Past

Ovarian cancer has been one of the most unique and perplexing diseases to diagnose and treat over the past 50 years. This is largely due to the inability of investigators to pinpoint its origins as well as the difficulty in detecting the tumor at a curable stage, combined with eventual resistance to chemotherapy [1]. Unlike endometrial and cervical malignancies which are preceded by recognizable precursor lesions in the respective site, the majority of ovarian carcinomas (high-grade serous carcinomas—HGSC) have been assigned to an ovarian origin by default. Systems for designating origin have been primarily dependent upon the location of tumor rather than the identification of a specific carcinogenic sequence [2]. Because most of these carcinomas do involve the ovarian surface, an origin in the ovarian surface epithelium was presumed [2]. For tumors that did not involve the ovarian surface or demonstrated scant evidence of ovarian involvement, a source in the peritoneal cavity was presumed. These so-called “primary peritoneal carcinomas” were generally held to arise within epithelial rests such as endosalpingiosis, endometriosis or other components of the “secondary Mullerian system” [2].

## 2. The Paradigm Shift

Several unrelated but temporally linked observations facilitated the emergence of a potential new source for HGSC. The first was the discovery of the *BRCA* cancer susceptibility genes [3]. The ability to identify patients at risk by testing for germline *BRCA1* and *BRCA2* mutations accelerated the adoption of risk-reducing salpingo-oophorectomies. This in turn increased the likelihood that early cancers would be discovered in the ovaries or fallopian tubes. A second observation arose from a concurrent study that underscored the rarity of early HGSC in the ovary [4]. A third observation was the progressive realization that both serologic screening and ultrasound demonstrate very little efficacy in detecting these HGSCs at a curable stage [5].

The most compelling evidence that suggested moving the origin of this tumor away from the ovary and to the fallopian tube arose around the year 2000, when investigators reported early serous carcinomas in the fallopian tubes of women with germline *BRCA1* or *BRCA2* mutations [6]. This was followed by a series of confirmatory reports identifying either serous cancers or epithelial abnormalities containing *TP53* mutations in the fallopian tube [7,8]. Subsequently, the sectioning and extensive examination of the fimbria (SEE-FIM) dissection protocol the distal fallopian tube, which is where the majority of early malignancies were found (Table 1) [9]. This was followed by studies of earlier precursor lesions in the fallopian tube, ranging from small stretches of epithelium (p53 signatures) to proliferations termed serous tubal intraepithelial lesions in transition or simply, serous tubal intraepithelial lesions (STIL) [7,10,11,12]. Based on these observations, a serous carcinogenic sequence was assembled in the distal tube which began with a p53 signature and terminated in a serous tubal intraepithelial carcinoma (STIC), with serous tubal intraepithelial lesions displaying some but not all the features of STIC.

Application of the SEE-FIM protocols to carefully examine the tubes of women at risk for HGSC accelerated the percentage of early cancers attributed to the distal tube, approaching 100% in some studies [9,13]. The tubal theory of high-grade serous carcinogenesis was thus superimposed upon the prior literature and like most new models, it began as a simple paradigm in which a precursor-to-cancer evolution occurred in the tube, followed by dissemination of the peritoneal surfaces [14]. This explained the rather rapid emergence of a malignancy which began as an occult carcinoma in the fallopian tube and then rapidly became advanced once the tumor was disseminated to the peritoneum.

## 3. Unanswered Questions

The above serous carcinogenic model required a transition from precursor to cancer in the fallopian tube which led investigators to multiple conclusions. The first was the assumption that the metastatic carcinoma was launched from a primary malignancy or neoplasm in the fallopian tube. Encouraging this were observations that up to 75% or more of HGSC were involved with the fallopian tube in some manner [15]. This has led to a consensus (based on circumstantial evidence) concluding that any significant tubal involvement implied that the malignancy first developed in the tube [16]. In retrospect, this model might be overly simplistic as it is based exclusively on the physical distribution of the malignant tumor. If a serous tubal intraepithelial carcinoma could not be detected it was often attributed to the fact that the early cancer was either not sampled or was obliterated by the tumor [16,17]. Again, this approach was based upon a model with a single mechanism, in which malignancy developed first in the tube and then spread. Concurrent with these assumptions was a recommendation from a clinical perspective that opportunistic salpingectomy should be practiced whenever possible to reduce the risk of eventual HGSC [18]. Salpingectomy alone was also proposed for managing women with *BRCA* mutations. The latter strategy is currently under study, but with careful guidelines to minimize risk to the patients who insist on ovarian preservation [19].

A second and entirely different viewpoint coming from this work was informed by critical pathological observation. Based on multiple studies, the frequency of an intramucosal carcinoma in the fallopian tube in patients with *symptomatic* or advanced HGSC ranged from as low as 10% to as high as 60% [20]. Either estimate left a large percentage of HGSC in which a clear-cut early malignancy in the fallopian tube could not be identified. Proponents of other potential sites of origin pointed to the ovarian surface epithelium or the secondary Mullerian system [21,22,23]. These proposals did not exclude the fallopian tube as the ultimate source of the tumor but theorized that benign cells derived from either the fallopian tubes or embryonic Mullerian rests (endosalpingiosis) in the pelvis (secondary Mullerian system) could undergo malignant transformation and explain the absence of a concurrent STIC. Moreover, the notion that STICs were invariably primary neoplasia of the tube, serving as launch pads for HGSC, was challenged when studies revealed that STICs could be secondary deposits from a widespread cancer rather than a primary site [24,25].

The variable frequency of primary cancerous lesions in the fallopian tube in women with HGSCs could be viewed from multiple perspectives. One proposal was that carcinomas emerging elsewhere, such the ovarian surface, could do so rapidly and thus bypass an obvious precursor lesion [22]. The observation that many cancers were associated with large cystic masses could be interpreted to mean that the lesions develop within pre-existing cysts such as endosalpingosis, endometriomas or adenofibromas [23]. However, there was little direct evidence (except in rare instances) that HGSCs arise from these preexisting conditions [26]. Two studies performing an exhaustive examination of the ovarian cortical inclusion cysts of women with *BRCA* mutations failed to identify any evidence of a precursor with *TP53* mutation [27,28]. Some authors proposed that different morphologic growth patterns might be more likely to be associated with a STIC. What these studies did show was that the overall frequency of a STIC in cases of advanced serous cancer cases from women with *BRCA* mutations was low, further emphasizing the likelihood that another pathway must be considered [29]. Another piece of information suggesting more than one mechanism of carcinogenesis was the similarity in mean age between women with *BRCA* mutations who presented with isolated serous tubal intraepithelial carcinoma and women with symptomatic widespread malignancy [30]. Given the assumption that there must be a lag period between the onset of early cancer and the later disseminated malignancy, the similarity in age between early and advanced *BRCA* mutation-associated HGSC was at odds with a more traditional precursor-cancer model [31]. Finally, molecular genetic studies of HGSC with and without an associated STIC failed to show any noticeable difference between the two groups [32]. In all of these tumors a *TP53* mutation was a requirement for precursor identification and in contrast to the fallopian tube, there was no consistent evidence of early or occult *TP53* mutations observed within alternate sites of origin, including endosalpingosis, endometriosis, or ovarian inclusion cysts, apart from sporadic reports [33,34].

In summary, it was clear that a very high percentage of early serous cancers discovered in asymptomatic women arose from within the fallopian tube. At the same time, a rather low percentage of advanced cancers were associated with a tubal malignancy. However, both cancer groups were genetically similar and there was no compelling evidence for an alternate origin containing a cancer precursor with a *TP53* mutation. This raised the fundamental question of how the fallopian tube could be involved in the development of HGSC in the absence of a STIC [35].

## 4. Serous Cancer Precursors in the Fallopian Tube

As mentioned above, starting around year 2000, investigators noticed that early serous cancer precursors could be found within the fallopian tubes. The exact carcinogenic sequence in which these putative precursors ultimately evolve into HGSC remains to be further elucidated, but it is clear that clonal *TP53* mutations in regions of the *TP53* gene known to be altered in HGSC are commonly detected in the distal fallopian tubes. In one study, fallopian tubes from 50% of controls and 70% of women with *BRCA* mutations contained at least one small stretch of 12 or more cells with a *TP53* mutation [36]. These small foci, termed p53 signatures by Lee et al., occurred in non-ciliated cells, similar to their malignant counterparts which are localized to the distal fallopian tube. These foci were associated with evidence of DNA damage supported by gamma-H2AX immunostaining [10]. This and other studies showed examples where these early lesions could be found in continuity with intraepithelial carcinomas, or occasionally malignancies in the fallopian tube [37]. Given the frequency of these lesions, they were presumed to be at very low risk for giving rise to an eventual serous cancer. Proliferative lesions, termed either tubal intraepithelial lesions in transition or serous tubal intraepithelial lesions (STIL) were less common and were considered an intermediate step between the p53 signature and STIC [10,12]. For the purposes of this discussion, this spectrum of tubal precursors containing *TP53* mutations will be designated as early serous proliferations or ESPs [38].

## 5. Early Serous Proliferations and “Precursor Escape”

Until recently, ESPs were not considered to play a major role in the development of HGSC. Given that they are found in up to 50% of the general population with low risk of HGSC, the overall risk for developing HGSC in an individual with one of these lesions would seem to be very low. Moreover, it is well known that women with *BRCA* mutations who present with an asymptomatic STIC carry only an approximate 5%-risk of ever developing a metastatic serous cancer [39]. Finally, the epidemiology of putative precursors to HGSC does not match their malignant counterpart, signifying gaps in our knowledge of the precursor-cancer connection [40,41]. Nevertheless, the only universally appreciated precursor candidate to date with a *TP53* mutation has been found in the fallopian tube [35].

In a recent study, Soong et al. performed meticulous sectioning of 32 consecutive cases of fallopian tubes from women with HGSC who did not have evidence of an early malignancy in the tubes [38]. The purpose of this study was to determine whether ESPs were present; and if they were, whether they share similar *TP53* mutations with the metastatic carcinomas. In this study, complete sectioning actually revealed an additional occult tubal malignancy in 3 of the 32 cases. However, the most striking finding was that in 12 cases, ESPs were identified with *TP53* mutations and in 9 of those 12 cases a *TP53* mutation was detected that was identical to the concurrent metastatic serous carcinoma. This study, for the first time, provided evidence of lineage continuity between ESPs and widespread HGSCs in the absence of a recognizable STIC or other early malignancy [38]. Based on this it appears that a small or early genetic lesion in the fallopian tube could be ultimately responsible for a later emergence of widespread serous carcinoma.

## 6. De-Mystifying a Paradox

The possibility that an early serous proliferation could escape the fallopian tube and ultimately emerge as a widespread intra-abdominal malignancy has the potential to resolve, at least in part, the decade-long question of why early and advanced HGSCs differ in association with STIC. The answer may be a dualistic model in carcinogenesis. In the classic scenario, a tubal serous carcinoma develops in the distal fallopian tube and over time invades or spreads to the ovaries and peritoneal surfaces, or lymph nodes, after which the tumor becomes more widespread (Figure 1) [42]. The prototype for this part of the model is the presence of STIC encountered in risk-reducing surgeries of women with *BRCA* mutations [8,12]. Approximately 5–10% of these patients will eventually manifest with a metastatic HGSC [43,44]. In the second part of the model (precursor escape), cells exfoliated from an ESP are maintained in some manner in the peritoneal cavity and in rare instances evolve into a malignancy which due to its location spreads rapidly throughout the peritoneal cavity, including the involvement of the ovarian surfaces and a retrograde spread to the tubes (Figure 2). Practically speaking, both parts of the model are a form of “precursor escape”, the first precursor being STIC and the second an ESP. However, this dualistic model explains the dual presentations: (i) The tumors that can be assigned to the tube and classified as “tubal” carcinomas are those that develop in the tube proper and will usually be discovered unexpectedly following surgical procedures for other disorders or following risk-reducing procedures. (ii) Those HGSCs that cannot be easily assigned to the tube will, by definition, be advanced, and might have arisen and spread in the peritoneal cavity. The potential for nonmalignant but genetically altered Mullerian epithelial cells originating from the tube to exist in the peritoneal fluid, or to travel from one site to another is plausible, particularly when one considers the pathogenesis of endometriosis [45].

## 7. Unanswered Questions

The promise of “precursor escape” or a similar mechanism is the possibility that it will settle the question of HGSC origin in those cases that do not manifest with a conspicuous “launch pad” in the tube, i.e., a STIC. If precursor escape is proven to be an important component of serous carcinogenesis, other existing models must be more critically reevaluated. Neither the ovarian surface epithelium/cortical inclusion cyst nor the secondary Mullerian system model is strongly supported by a recognizable precursor with a *TP53* mutation [23,24]. Moreover, if STICs and ESPs occasionally possess the capability to persist and reemerge as metastatic HGSCs, it is entirely conceivable that less conspicuous but genetically similar lesions could contribute to this disease. This possibility must be excluded by rigorous pathological and molecular analysis of the fallopian tubes of women with HGSC to establish the extent of lineage and relationship between these early proliferations and advanced malignancy. If proven, this association will certainly inform future strategies for serous cancer prevention, both in the general population and in women at high risk for this disease. In the process of this research a more complete understanding might be gained of previously unappreciated biological events that could be taking place during the occult phase of serous cancer development [46,47].

## 8. Conclusions

A nagging question regarding pelvic serous carcinogenesis has been a paradox created by the fact that when HGSC is detected “early”, it is invariably confined to the fallopian tube; but when advanced, it often does not reveal an obvious malignant launch point in the fimbrial mucosa. We have postulated previously that the most likely explanation for the latter scenario would be either an occult origin in the pelvic ovarian surface epithelium or an alternate fallopian tubal precursor [10,48]. The concept of “early precursor escape” revises both of these hypotheses by postulating an alternate precursor in the tube (e.g., ESP) that matures into an “advanced” malignancy on the peritoneal surfaces. If this pathway is corroborated by further study, the elusive question of why many HGSCs are not associated with a STIC will be answered. Moreover, it will provide a framework for assessing other risk factors to HGSC that influence not only the genesis of ESPs but more importantly, their transformation in the peritoneal cavity.

## Figures and Tables

**Figure 1 cancers-10-00468-f001:**
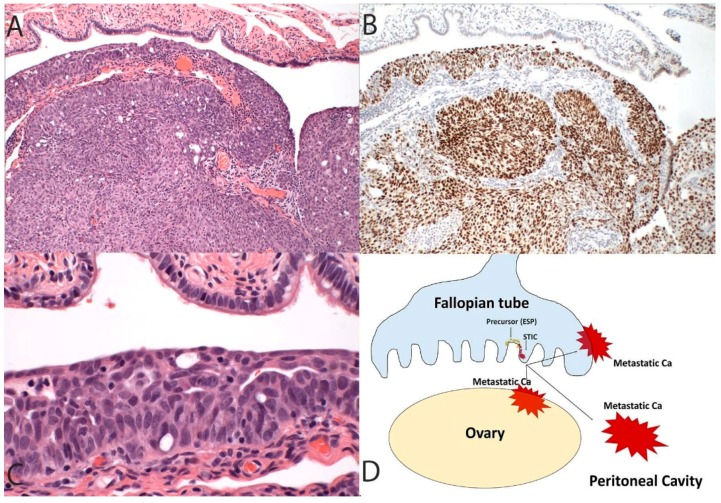
In this more conventional model of high-grade serous cancer development, the tumor initiates in the fallopian tube (**A**, 100×) and contains a *TP53* mutation highlighted in this case by diffuse immunostaining for p53 (**B**, 100×) as well as a recognizable serous tubal intraepithelial carcinoma (**C**, 400×). The schematic in (**D**) depicts a primary tumor of the tube that spreads to the ovary and peritoneal cavity. This progression sequence is one that might be impacted, albeit in a limited way, by early detection schemes. It is also the sequence that, when discovered early, will invariably implicate the fallopian tube as the source of the malignancy.

**Figure 2 cancers-10-00468-f002:**
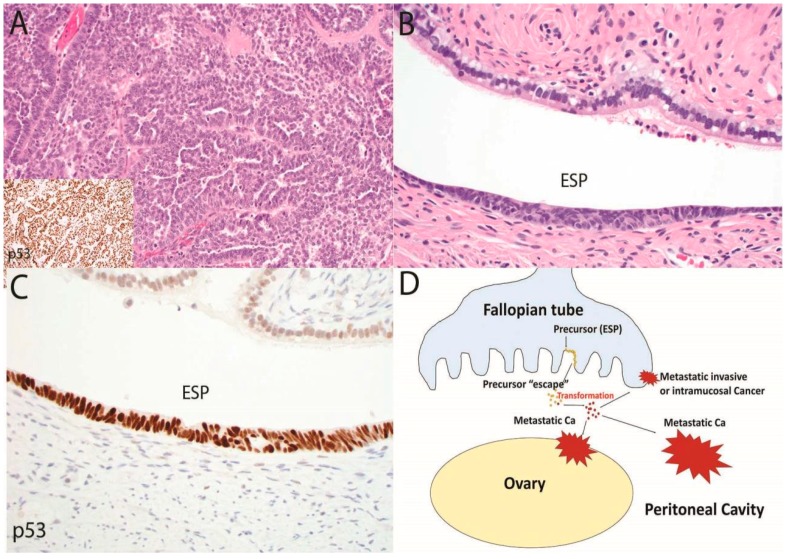
In this second and complementary model of serous tubal carcinogenesis, an advanced malignancy with a *TP53* mutation is present (**A**, 100×), yet the only histologic abnormality that can be discerned in the fallopian tube is an early serous proliferation (ESP) (**B**, 400×) with strong p53 staining indicating a *TP53* mutation (**C**, 400×). In this scenario, summarized in (**D**), a typical serous tubal intraepithelial carcinoma may not develop; rather cells may exfoliate from the ESP via “precursor escape” and ultimately emerge as an advanced malignancy. This sequence explains the discovery of widespread serous cancer in the absence of an obvious tubal mucosal malignancy. In contrast to the sequence depicted in Figure 1, the initiation of malignancy in the peritoneal cavity would soon be followed by widespread disease that is nearly impossible to intercept in its early stages.

**Table 1 cancers-10-00468-t001:** Sectioning and extensively examining the fimbria (SEE-FIM) protocol [9].

1	Fix the fallopian tubes for 2 h.
2	Amputate the distal third and thinly (1 mm intervals) section in a sagittal plane (longitudinally) to gain the maximum exposure of the mucosa to histologic review.
3	Section the remainder of the tube at 1 mm intervals.
4	Submit the entire tube for histologic review if the patient is suspected to be at higher risk for high-grade serous carcinomas (HGSC) or if the patient has a concurrent HGSC, other uterine or extra-uterine Mullerian epithelial malignancy.
5	In routine surgical cases, submit the distal fallopian tube as appropriate.

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
