# Peer review of "Back to the Future? The Fallopian Tube, Precursor Escape and a Dualistic Model of High-Grade Serous Carcinogenesis"

_cancers, 2018, doi:10.3390/cancers10120468_

Reviewer 1 Report

This is a well-written manuscript focusing on the emerging role of fallopian tube as a possible origin of ovarian cancer. Some comments are given to the authors to improve their paper: 

1-There are some very nice papers from the leader Robert Kurman working on this problem that the authors didn't discuss. Please check these papers below and add some paragraphs based on the data discussed in these reviews:

https://ajp.amjpathol.org/article/S0002-9440(16)00008-0/fulltext

https://journals.lww.com/ajsp/fulltext/2010/03000/The_Origin_and_Pathogenesis_of_Epithelial_Ovarian.18.aspx

https://journals.lww.com/ajsp/fulltext/2010/03000/The_Origin_and_Pathogenesis_of_Epithelial_Ovarian.18.aspx

There is also a very nice review published in Nature Reviews Cancer that you may use:

https://www.nature.com/articles/nrc3144

2-In the "Precursor Escape?" section:

From "In a recent study, Soong et al performed meticulous" to " an early malignancy in the tubes." A reference must be added here.

3-The discussion provided by the authors is much appreciated but in various paragraphs, it needs references!!.

4- Please increase the size and the quality of the figures (TIFF is required). They are not visible enough to check the histology !.

5-"Figure 2" footnotes, from "This sequence explains" to "depicted in Figure 1, this sequence should ".  Should what ??!

6-I will appreciate if the authors add a box or table/figure about the SEE-FIM protocol for identifying the fallopian tube precursor lesions so the readers will be informed about its importance in modern pathology.

7- There is a growing evidence of the role of hemoglobin the pathogenesis of ovarian carcinomas. Please add a paragraph or a chapter about this phenomenon. You will find some nice data here: 

https://onlinelibrary.wiley.com/doi/full/10.1002/path.4875

8-In the whole manuscript: 

The standard convention for human and animal genes and proteins is:

Human - upper case, italics for genes and Roman font for proteins

Animal - lower case, italics for genes and Roman font for proteins

Please check/Correct. Moreover, abbreviations must be defined at first use and used thereafter. 

Author Response

1-There are some very nice papers from the leader Robert Kurman working on this problem that the authors didn't discuss. Please check these papers below and add some paragraphs based on the data discussed in these reviews:

We are happy to consider publications by the reviewer's favorite author. Bob's review in Am J Path fits nicely in the discussion and will be cited along with some others.

2-In the "Precursor Escape?" section:

From "In a recent study, Soong et al performed meticulous" to " an early malignancy in the tubes." A reference must be added here.

Reference is cited.

3-The discussion provided by the authors is much appreciated but in various paragraphs, it needs references!!.

Additional references are cited.

4- Please increase the size and the quality of the figures (TIFF is required). They are not visible enough to check the histology !.

TIFF files will be uploaded.

5-"Figure 2" footnotes, from "This sequence explains" to "depicted in Figure 1, this sequence should ".  Should what ??!

Addressed

6-I will appreciate if the authors add a box or table/figure about the SEE-FIM protocol for identifying the fallopian tube precursor lesions so the readers will be informed about its importance in modern pathology.

We added a box for this.

7- There is a growing evidence of the role of hemoglobin the pathogenesis of ovarian carcinomas. Please add a paragraph or a chapter about this phenomenon.

We will see if we can fit this in. I assume the reviewer would like their commentary as well as the original manuscript cited.

8-In the whole manuscript: 

The standard convention for human and animal genes and proteins is:

Human - upper case, italics for genes and Roman font for proteins

Animal - lower case, italics for genes and Roman font for proteins

Please check/Correct. Moreover, abbreviations must be defined at first use and used thereafter

Will do

Reviewer 2 Report

The author demonstrated dualistic carcinogenetic model of high-grade serous type of ovarian canecr with"precursor escape" theory. This idea is elegant and raises a new question for us about conventional knowledge of ovarian serous carcinogenesis.

#In Fig 1 and2: Alphabets in these Figures (A, B, C,,,) are unclear. Please improve them.

#In Fig 2 legend may be incomplete in the square.

#In Fig 1 and 2: The author should enlarge “D” figures to make us easily understand.

Author Response

#In Fig 1 and2: Alphabets in these Figures (A, B, C,,,) are unclear. Please improve them.

Corrected the text

#In Fig 2 legend may be incomplete in the square.

Addressed

#In Fig 1 and 2: The author should enlarge “D” figures to make us easily understand

We uploaded TIFF figures that can certainly be enlarged on-line as the publisher sees fit.

Reviewer 3 Report

In this review, authors reviewed the literature on the origin of HGSC, which is a controversial topic in this field. However, authors proposed two different models to explain how HGSC arises from fallopian tubes including a conventional model that how ESPs spread tumor to ovaries and peritoneal cavity through STIC and second model that ESPs undergo escaping and transformation to spread tumor to ovaries and peritoneal organs. Although more experimental evidence still is needed to verify this model, but prompts to think a new strategy in ovarian cancer prevention and therapy.

Few minor issues:

1.      In Figure 2, figure legend is not fully displayed.

2.      In line6 of page 5, there is a reference missing.

3.      Figures should have scale bar or magnifications.

4.      Sections should also be stained with Ki67 or PCNA to show epithelial cell proliferation in addition to p53.

Author Response

1.      In Figure 2, figure legend is not fully displayed.

Fixed

2.      In line6 of page 5, there is a reference missing.

Fixed

3.      Figures should have scale bar or magnifications.

Addressed

4.      Sections should also be stained with Ki67 or PCNA to show epithelial cell proliferation in addition to p53.

This was not done for this particular case due to a lack of additional material. We do not feel that the proliferative index need to illustrated to make the point. The lesion is simply illustrated as an example of a process that does not meet the criteria for STIC.

Round  2

Reviewer 1 Report

None